# DNA Penetration into a Lysozyme Layer at the Surface of Aqueous Solutions

**DOI:** 10.3390/ijms232012377

**Published:** 2022-10-16

**Authors:** Nikolay S. Chirkov, Shi-Yow Lin, Alexander V. Michailov, Reinhard Miller, Boris A. Noskov

**Affiliations:** 1Institute of Chemistry, Saint Petersburg State University, 198504 St. Petersburg, Russia; 2Department of Chemical Engineering, National Taiwan University of Science and Technology, 43, Keelung Road, Section 4, Taipei 106, Taiwan; 3Physics Department, Technical University Darmstadt, Hochschulstraße 8, D-64289 Darmstadt, Germany

**Keywords:** DNA, lysozyme, adsorption kinetics, dilational surface rheology, Langmuir monolayers

## Abstract

The interactions of DNA with lysozyme in the surface layer were studied by performing infrared reflection–absorption spectroscopy (IRRAS), ellipsometry, surface tensiometry, surface dilational rheology, and atomic force microscopy (AFM). A concentrated DNA solution was injected into an aqueous subphase underneath a spread lysozyme layer. While the optical properties of the surface layer changed fast after DNA injection, the dynamic dilational surface elasticity almost did not change, thereby indicating no continuous network formation of DNA/lysozyme complexes, unlike the case of DNA interactions with a monolayer of a cationic synthetic polyelectrolyte. A relatively fast increase in optical signals after a DNA injection under a lysozyme layer indicates that DNA penetration is controlled by diffusion. At low surface pressures, the AFM images show the formation of long strands in the surface layer. Increased surface compression does not lead to the formation of a network of DNA/lysozyme aggregates as in the case of a mixed layer of DNA and synthetic polyelectrolytes, but to the appearance of some folds and ridges in the layer. The formation of more disordered aggregates is presumably a consequence of weaker interactions of lysozyme with duplex DNA and the stabilization, at the same time, of loops of unpaired nucleotides at high local lysozyme concentrations in the surface layer.

## 1. Introduction

The interaction of nucleic acids with proteins is essential for all known forms of life on Earth and has been continuously studied since the first half of the 20th century. The multistep process of gene expression in living organisms is impossible without the formation of DNA complexes with various proteins. DNA is a polyanion at pH values close to 7; therefore, it is able to interact electrostatically with positively charged proteins in this range [1]. A typical example of such interactions is the formation of chromatin, a compact complex of DNA with histones [2]. At the same time, DNA can also interact with negatively charged proteins, for example, with serum albumins [3,4,5], mainly due to the non-uniform charge distribution along the surface of a protein globule [6].

A notable part of the extensive literature on the interaction of DNA with various proteins is dedicated to the formation of DNA/protein structures as solid substrates [7,8,9]. These studies are motivated by the importance of protein assembly onto DNA strands for both basic research and potential applications in nanotechnology [8]. Special interest is generated by network structures of DNA/protein complexes. For example, DNA/histone complexes, which can be formed upon adsorption onto the mica surface [9]. It has recently been demonstrated that such networks can possess antimicrobial properties, depending on their composition [10]. It is unclear, however, if such structures can be formed at the liquid–fluid interface. Paul et al. studied a spread layer of Cytochrome c/λ-DNA complex and discovered the presence of fibrous aggregates using atomic force microscopy (AFM) [11].The size of these aggregates was greater than that of λ-DNA, and they tended to merge into larger rods at increased incubation times. At the same time, no well-defined network of the aggregates has been observed.

Lysozyme is a well-known protein with antimicrobial properties [12]. It is also a basic protein with an isoelectric point at about pH 11 [13], making it a suitable model for the investigation of DNA–protein interactions at the solution–air interface. It is well-established that DNA–lysozyme complex formation is driven predominantly by electrostatic interactions [14,15,16,17,18]. The morphology of DNA–lysozyme aggregates depends strongly on the molar ratio between the components. Lundberg et al. observed the formation of DNA/lysozyme flexible worm-like assemblies at low lysozyme to DNA molar ratios [18]. Zhang et al. have recently demonstrated that the binding of lysozyme globules to DNA can result in the formation of both “over-” and “undercharged” complexes with a different morphology due to electrostatic interactions governed by the protein concentration [17]. Very recently, Morimoto et al. studied the influence of high lysozyme concentrations under macromolecular crowding conditions, in [19]. Their study proved that at high concentrations, lysozyme stabilizes mainly non-canonical structures of DNA, in particular, loops comprising unpaired nucleotides but not DNA duplexes.

It has been shown recently that surface rheology can be applied to investigate the formation mechanism of DNA-containing nanostructures at the solution–air interface [20,21,22]. This technique together with optical methods made it possible to study the penetration of DNA into a monolayer of a synthetic polyelectrolyte poly(N,N-diallyl-N-hexyl-N-methylammonium) chloride (PDAHMAC) and to elucidate the mechanism of a regular DNA/polyelectrolyte network formation at the liquid–gas interface [22]. In this work, the same multi-technique approach is applied to investigate DNA penetration into a lysozyme layer spread onto an aqueous subphase. The main aim is to elucidate the peculiarities of DNA/protein interactions at the liquid–gas interface and to verify the possibility of a network formation. In particular, we aim to estimate the DNA/lysozyme aggregation in a crowded molecular environment in the lysozyme spread layer.

## 2. Results

### 2.1. Surface Dilational Rheology

The kinetic dependencies of the surface pressure and surface elasticity can provide useful information on conformations of macromolecules in the surface layer [23], in particular, in cases of DNA/surfactant [20] and DNA/polyelectrolyte [22] systems. The case of the DNA/lysozyme system proved to be more complicated. Spreading of a concentrated lysozyme solution onto the surface of a buffer solution produces a layer with relatively high dynamic elasticity, close to the data of lysozyme adsorption layers [24]. The subsequent injecting of DNA into the solution under the protein layer does not lead to noticeable changes of the surface elasticity, unlike DNA injection under a layer of a synthetic cationic polyelectrolyte (Figure 1a) [22]. All the changes were close to the error limits, even 15 h after DNA injection. A similar trend holds true for kinetic dependencies of the surface pressure (Figure 1b). There is only a slight increase in the surface pressure 15 h after the DNA injection if the initial surface pressure of the lysozyme layer is 10 mN/m.

Nevertheless, DNA penetration into the lysozyme layer can be detected by means of compression isotherms. One hour after the lysozyme spreading, an almost linear increase in surface pressure was observed due to the interaction between rigid lysozyme globules, without any clear phase transition regions (Figure 2). The lysozyme spreading onto the surface of a buffer solution with a subsequent DNA injection and 1 h incubation time did not change the shape of the isotherm. At the same time, it leads to a noticeable shift of the isotherm to larger surface areas, which is a well-known indicator of DNA penetration into the spread layers of different compositions [25].

### 2.2. Atomic Force Microscopy

AFM data provide an additional argument in favor of the DNA incorporation into the lysozyme layer. When no DNA is injected into the subphase and lysozyme is transferred onto the mica surface at a surface pressure of 10 mN/m after incubation for ten hours, one can observe a continuous layer with some inhomogeneities, which are most likely related to the shortcomings of the transfer/drying process (Figure 3a). The injection of DNA into the subphase noticeably changes the micromorphology of the layer. At a surface pressure of 10 mN/m and at 10 h after DNA injection, one can observe a number of extended and interconnected threads, surrounded by smaller aggregates (Figure 3b). These threads are 3 to 4 nm in the Z-direction, and thereby are slightly thicker than a double stranded DNA molecule with a diameter of ~2 nm. A similar increase in the thickness was also observed for the Cytochrome c/DNA system [11], which can be attributed to the formation of DNA/lysozyme complexes in the surface layer. This observation and the absence of threadlike aggregates on the mica surface if there is no DNA in the subphase provide further evidence of DNA penetration into the lysozyme layer at long surface lifetimes, despite only weak changes in the dynamic surface properties (Figure 1). The further increase in the surface pressure of transfer (up to 20 mN/m) leads to the formation of a thick heterogeneous layer, consisting of large folds or ridges, embedded in a dense layer (Figure 3c). These folds can contain DNA, or may be a result of the lysozyme multilayer formation. At even higher surface pressures of transfer, these folds merge and form even larger aggregates, although the main features of the layer morphology do not change (Figure 3d).

### 2.3. Ellipsometry

The ellipsometric results also indicate DNA penetration into the lysozyme layer. The spreading of a lysozyme solution onto the buffered subphase leads to a substantial increase in the Δ_s_ value (about 2 degrees, Figure 4). The subsequent 2-fold surface compression increases this value up to 3 degrees without further noticeable changes, thereby indicating the formation of a stable close-packed film. The further 4- and 8-fold compressions lead to quick relaxations of the Δ_s_ values, presumably due to the lysozyme dissolution in the subphase. At the same time, Δ_s_ does not reach the value before the corresponding compressions, and amounts to around 5.3 degrees after the relaxation at 8-fold compression.

DNA injection leads only to a slight increase in Δ_s_ at 2.5 h after the injection (Figure 4), which is consistent with measurements of the dynamic surface properties (Figure 1). The subsequent 2-fold compression shows a strong influence of DNA on the changes of the ellipsometric signal after compression. In particular, a 2-fold compression of the lysozyme layer on a DNA-containing subphase and an 8-fold compression of the lysozyme layer on the subphase without DNA lead to almost the same values of the ellipsometric angle. Δ_s_ reaches even higher values at the subsequent 4- and 8-fold compressions. In the latter case, Δ_s_ reaches approximately 13.5 degrees and decreases slowly after that to about 10.8 degrees. The obtained results show not only the DNA penetration into the lysozyme layer, but also the influence of DNA on the layer structure and strong interactions of the components in the layer decreasing the protein dissolution after compression.

### 2.4. IRRAS

IRRAS facilitates the separation of the signals from different components of the adsorption layer. Before DNA injection, three bands were seen in the IRRAS spectrum of the lysozyme layer: one narrow band close to 1250 cm^−1^ (Figure 5a), and two wide bands in the region of 1400–1800 cm^−1^ (Figure 5b). The narrow band close to 1250 cm^−1^ is the component of the amide III band; the two broad bands are the amide II and amide I bands. All of the bands appear due to the peptide bond vibrations in the lysozyme layer [26]. DNA injection into the subphase results in the emergence of two bands at around 1085 and 1225 cm^−1^, corresponding to the symmetric and asymmetric stretching of the phosphate groups of DNA, respectively [27]. The intensity of the band at around 1085 cm^−1^ increases with the surface age after DNA injection without a noticeable induction period, unlike the case of DNA penetration into a monolayer of poly(N,N-diallyl-N-hexyl-N-methylammonium) chloride (PDAHMAC) [22], which reaches a maximum value in approximately 2.5 h, and changes only within error limits after that. The intensity of the weak 1225 cm^−1^ band is much less than that of the 1085 cm^−1^ band and is close to the error limits before surface compression (Figure 5a), but exceeds them slightly after compression (Figure 6a). Note that DNA penetration into the lysozyme layer increases the intensity of both the amide II and amide I bands of lysozyme as well, presumably due to the redistribution of lysozyme molecules in the system under the influence of adsorbing DNA strands (Figure 5b).

The 2-fold surface compression leads to a gradual increase in the intensity of both DNA phosphate bands (Figure 6a). The intensity of the bands reaches constant values approximately 2 h after compression. This period can correspond to a relatively slow reorganization of the surface layer structure. Note that the intensity of these bands is not reduced upon compression, thereby indicating that DNA is not expelled into the subphase and the large folds, which are observed by AFM at high surface pressures; they are caused by DNA molecules surrounded by the protein in the layer (Figure 3c,d). The intensity of amide II and amide I bands of lysozyme after surface compression reaches constant values faster than the intensity of the DNA phosphate bands, presumably due to the greater mobility of lysozyme molecules in the surface layer compared to DNA strands (Figure 6b). This increase in the intensity of spectrum bands corresponds to the formation of a thick lysozyme layer, as can be observed in the AFM images (Figure 3c,d).

## 3. Discussion

It is known that the affinity of DNA to different polyions is a function of the persistence length, which characterizes the polyelectrolyte chain stiffness [6]. DNA is one of the stiffest polymers and has a persistence length of ∼50 nm in 0.1 M NaCl aqueous solution [28]. Lysozyme globules can be considered as ellipsoidal with approximate dimensions of 3.3 × 5.5 × 3.3 nm and a positive net charge of +8 at pH 7 [16]. Owing to the globules’ compact size compared to DNA molecules, they can easily bind to DNA strands in a “beads on a string” manner. Such binding can lead to the collapse of DNA strands with the formation of compact complexes at certain protein concentrations [17]. At the same time, the AFM data of this study show long DNA threads at the liquid surface (Figure 3b). The threads seem to be covered by protein molecules to some extent. The high local concentration of the protein in the surface layer can lead to the formation of “overcharged” DNA–lysozyme complexes. In this case the electrostatic repulsion between protein molecules bound to DNA strands can prevent DNA compaction, which is observed in the bulk phase.

Although the compression isotherms, AFM, ellipsometry, and IRRAS results clearly demonstrate that DNA can interact with lysozyme at the solution–air interface, the corresponding changes of the dynamic surface properties are not far from the error limits.

The dilational surface elasticity is a surface property, which is especially sensitive to the structure of a layer of macromolecules at the liquid–gas interface [20,29,30]. While the formation of a network of fibrous DNA/PDAHMAC aggregates leads to drastic changes in the surface elasticity and a slighter variation of the dynamic surface tension [22], the obtained results show that the DNA/lysozyme interactions in the surface layer do not lead to noticeable changes of the surface elasticity and the formation of a rigid network (Figure 1a), although some interconnected DNA fibrils are clearly visible in the AFM images (Figure 3b).

The properties of PDAHMAC monolayers are similar to those of monolayers of conventional insoluble surfactants on an aqueous subphase with a two-dimensional phase transition from a liquid-expanded layer to a condensed layer [22]. The condensed phase is presumably formed as a result of cohesion between the hexyl groups of the polymer at the boundary of the monolayer with the air phase. On the other hand, all the monomers of PDAHMAC can form cations in water leading to a relatively high surface charge density of the PDAHMAC monolayer. These characteristic features presumably lead to the formation of a regular network of fibrous DNA/PDAHMAC aggregates after DNA injection beneath the polyelectrolyte monolayer. A similar network is formed at the interface in mixed solutions of DNA with soluble cationic surfactants [20].

The lysozyme layer differs significantly from the PDAHMAC layer. Firstly, lysozyme has both cationic and anionic groups on the surface of globules, although its total charge is positive at a pH close to 7. As a result, the surface charge density of the lysozyme layer is lower than for a layer of a synthetic polyelectrolyte. Secondly, the lysozyme globule is larger than a DAHMAC monomer; therefore, its layer at the liquid surface is rougher than that of PDAHMAC. Thirdly, the hydrophobic and hydrophilic groups are separated in a PDAHMAC monolayer, but this is less feasible for the lysozyme layer due to a more complex molecular structure. All these distinctions from layers of the synthetic polymer can result in weaker interactions of DNA molecules with the lysozyme layer, leading to the lack of a notable effect of DNA injection into the subphase on the properties of the lysozyme layer. Relatively weak interactions of lysozyme molecules with duplex DNA at high protein concentrations have been corroborated by a recent stability analysis of DNA oligonucleotide structures [19]. At the same time, it was found that basic globular proteins at high concentrations stabilized strongly as bulge loop structures. The non-canonical structures of DNA, e.g., loops comprising unpaired nucleotides, are quite probable in the investigated polydisperse DNA samples; in this case, the crowding of lysozyme results in strong interactions rather than with duplex DNA structures. The lysozyme interactions with loops of unpaired nucleotides presumably do not favor the formation of long strands of relatively small diameters but induce the formation of more disordered structures at a high protein concentration. The high lysozyme concentration in spread layers mimics intracellular crowding; therefore, the investigated system can model the effects of intracellular crowding on DNA behavior in living cells.

The ellipsometric and IRRAS results clearly indicate the formation of DNA/lysozyme complexes in the surface layer (Figure 4 and Figure 5). Although the surface activity of DNA is very low, it penetrates the lysozyme layer and strongly increases the signal of optical methods due to the binding of lysozyme molecules by DNA strands. The DNA/lysozyme complexes, however, are separated and do not form a continuous network in the surface layer, unlike the DNA/PDAHMAC complexes—as indicated by the surface rheological data (Figure 1).

The DNA adsorption into the system with a lysozyme layer is a fast process and presumably is controlled by the diffusion from the bulk phase. The intensity of the IRRAS bands corresponding to the symmetric and asymmetric stretching of the phosphate groups starts to increase 13 min after DNA injection as compared to the spectrum of a lysozyme layer (Figure 5). This characteristic time is one order of magnitude shorter than in the case of DNA penetration into a monolayer of a synthetic polyelectrolyte [22]. Moreover, DNA interaction with a PDADMAC monolayer requires a nucleation step, which is connected with the local partial destruction of the monolayer, to overcome steric hindrances for the electrostatic interactions between oppositely charged groups of DNA and the polyelectrolyte. On the contrary, the formation of DNA/lysozyme complexes is a continuous and much faster process without any indications of the nucleation process.

## 4. Materials and Methods

### 4.1. Materials

Calf thymus DNA (sodium salt, highly polymerized and polydisperse), lysozyme from chicken egg white and Trizma base from Sigma-Aldrich (Germany) were used as received. Sodium chloride from Vekton (St. Petersburg, Russia) was annealed in a furnace for 6 h at about 800 °C to eliminate organic impurities. All the solutions were prepared in triply distilled water.

### 4.2. Sample Preparation

DNA fibers were dissolved in a buffer solution (10 mM Tris plus 20 mM NaCl) adjusted to pH 7.6 with HC. This stock solution was stored at 4 °C for no longer than 2 weeks. The lysozyme layer was formed by spreading a 2 g/L lysozyme solution in Tris-HCl buffer onto the same buffer solution using a Hamilton microliter syringe (USA). The protein surface concentration after spreading was 1.5 × 10^−5^ g/cm^2^ or 1.5 × 10^−6^ g/cm^2^ if the compression isotherms were measured. The drops of 10-20 µL were placed successively on an inclined glass plate, installed onto the side of a Teflon Langmuir trough using a Teflon holder and partly immersed into the subphase. After that, an appropriate amount of DNA stock solution was injected into the subphase. The Langmuir trough and glassware were cleaned with a sulfochromic mixture and thoroughly washed and dried before use.

### 4.3. Methods

#### 4.3.1. Dilatational Surface Elasticity and Surface Tension

The dynamic dilatational surface elasticity was measured by the oscillating barrier method described elsewhere [22]. Periodic expansions/compressions of the liquid surface were created by a Teflon barrier moving back and forth along brims of a rectangular trough. The oscillation frequency and amplitude were kept constant at 0.1 Hz and 1.2%, respectively. The surface tension oscillations were measured with a rectangular Wilhelmy glass plate and sandblasted to ensure complete wetting. Before the spreading of the lysozyme solution, the subphase surface was cleaned by suction with a Pasteur pipette. All the measurements were carried out at 20 ± 1 °C.

#### 4.3.2. Ellipsometry

A Multiskop null ellipsometer (Optrel GbR, Germany) was used to determine relative changes of surface properties. All measurements were carried out at an angle of incidence close to the Brewster angle of water (49°). The light reflection from the interface changes the phase and amplitude of parallel and perpendicular components of elliptically polarized light. These changes are determined by the optical properties of the interface and characterized by two ellipsometric angles, a relative amplitude change ψ, and a relative phase shift Δ after reflection. These angles can be connected with the two complex reflection coefficients r_p_ and r_s_ by the following equation:$$\frac{r_{p}}{r_{s}}=\tan\left(\psi \right)e^{i\Delta }$$

The relation between the reflection coefficients, on the one hand, and the wavelength of the incident light, the angle of incidence, the refractive indexes of the bulk phases, the refractive index, and the thickness of the surface film, on the other hand, can be obtained within the framework of a specific model of the investigated system. In the basic model of a thin isotropic layer of uniform density, the difference in ellipsometric angles Δ_s_ between those of the investigated system (∆) and of the subphase (Δ_0_) is proportional to the adsorbed amount Γ [31].

#### 4.3.3. IRRAS

The IRRAS spectrometer was described elsewhere [20]. It consisted of a Nicolet 8700 FTIR instrument (Thermo Scientific, USA) and a Tabletop Optical Module (TOM). A linear wire grid polarizer was used to form the s- and p-polarization of the IR radiation (perpendicular to the plane of incidence and in the plane of incidence, respectively). The spectrometer and TOM were purged with nitrogen. In all spectra measurements, 2048 scans were collected at a 40° angle of incidence. The measurements were carried out for more than 10 h with a resolution of 4 cm^−1^ for both polarizations. IRRAS data are plotted as reflectance–absorbance (RA) versus wavenumber, RA = −log(R/R_b_), where R and R_b_ are the reflectivities from the system with a protein layer, with or without DNA, and pure buffer solutions, respectively.

#### 4.3.4. Surface Pressure-Area Isotherms

The compression isotherms were measured with a Langmuir film balance (KSV NIMA, Finland-Sweden). The surface compression was executed using two Teflon barriers moving along the polished brims of a Langmuir trough. The surface pressure was measured with the Wilhelmy plate method using a paper plate. The barrier speed was 15 cm^2^/min.

#### 4.3.5. Atomic Force Microscopy

The surface films of lysozyme and DNA/lysozyme were transferred from the liquid surface onto a freshly cleaved mica plate by performing horizontal dipping and dried in a desiccator at 4 °C; this was followed by an investigation with atomic force microscopy using NTEGRA Spectra setup (NT-MDT, Russia) in a tapping mode.

## 5. Conclusions

DNA penetration into a lysozyme layer spread onto an aqueous subphase was, to the best of our knowledge, studied here for the first time. The application of ellipsometry and IRRAS assisted in discovering an increase in the surface concentration of DNA molecules after DNA injection into a subphase, while the dynamic surface elasticity changed only slightly in this case. Despite the formation of DNA/lysozyme complexes within the surface layer, a regular continuous surface network is not formed and the DNA/lysozyme complexes are separated in the surface layer, unlike the case of DNA penetration into a layer of synthetic cationic polyelectrolyte. The interaction of a lysozyme layer with DNA molecules is a continuous process without a nucleation step, which was observed earlier in DNA/PDAHMAC systems [22]. The observed peculiarities of the properties of mixed DNA/lysozyme layers likely have two main causes. First, the interactions between the DNA and lysozyme layer are weaker than in the case of DNA penetration into PDAHMAC layers. Second, lysozyme interacts at high local concentrations in the surface layer mainly with non-canonical structures of DNA, leading to their stabilization and the formation of more disordered aggregates.

## Figures and Tables

**Figure 1 ijms-23-12377-f001:**
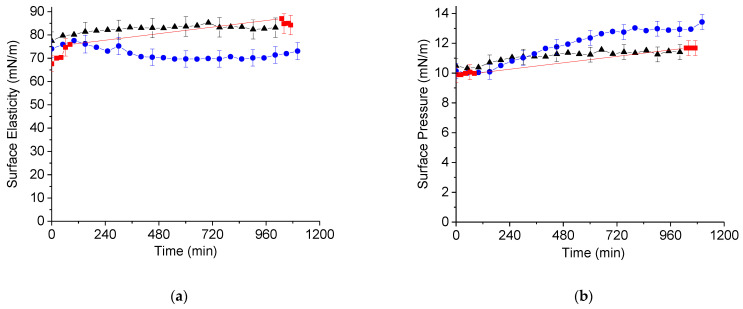
Kinetic dependencies of surface pressure (**a**) and elasticity (**b**) of lysozyme layers after a DNA injection into the subphase. The total DNA concentration is 14 μM, and the initial surface pressure is 10 mN/m. The three sets of experimental data correspond to different independent measurements. Lines are guides for the eye.

**Figure 2 ijms-23-12377-f002:**
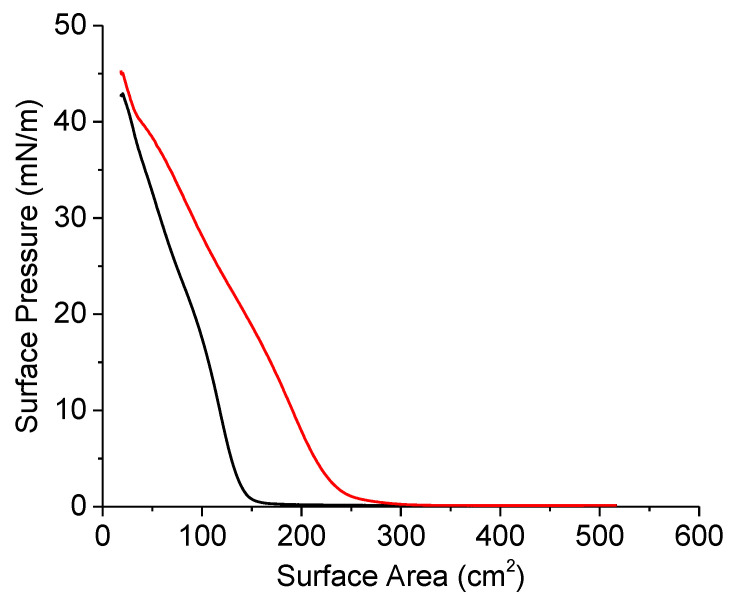
Compression isotherms of a lysozyme layer 1 h after spreading (black line) and at 1 h after spreading and a DNA injection (red line). The total DNA concentration is 50 μM.

**Figure 3 ijms-23-12377-f003:**
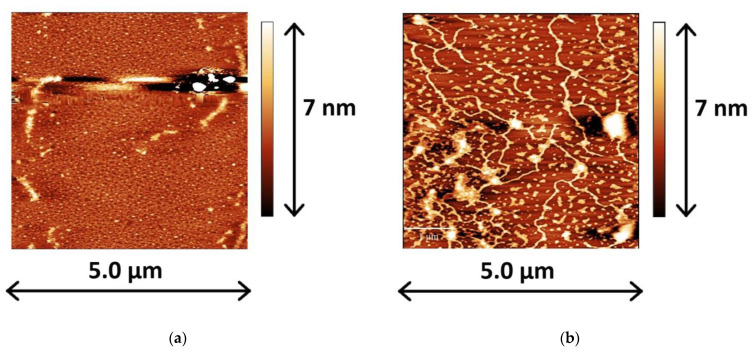
AFM images of a lysozyme layer transferred onto the mica surface at a surface pressure of 10 mN/m (**a**) and mixed lysozyme/DNA layers transferred onto a mica surface at surface pressures of 10 (**b**), 20 (**c**) and 30 (**d**) mN/m. The total DNA concentration is 50 μM.

**Figure 4 ijms-23-12377-f004:**
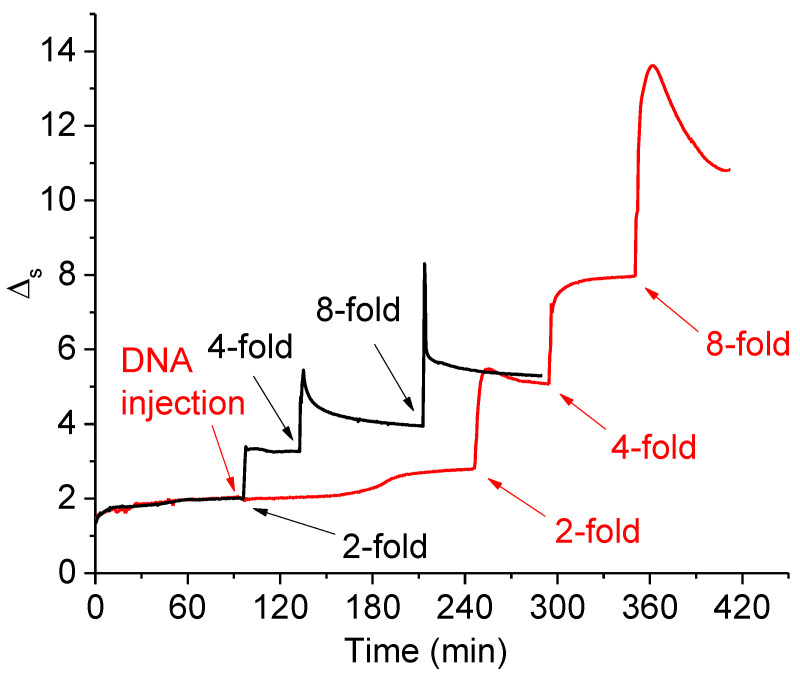
Kinetic dependencies of the ellipsometric angle Δs without (black line) and with (red line) DNA injection into the subphase. Arrows indicate the subsequent surface compressions and DNA injection. The total DNA concentration is 50 μM.

**Figure 5 ijms-23-12377-f005:**
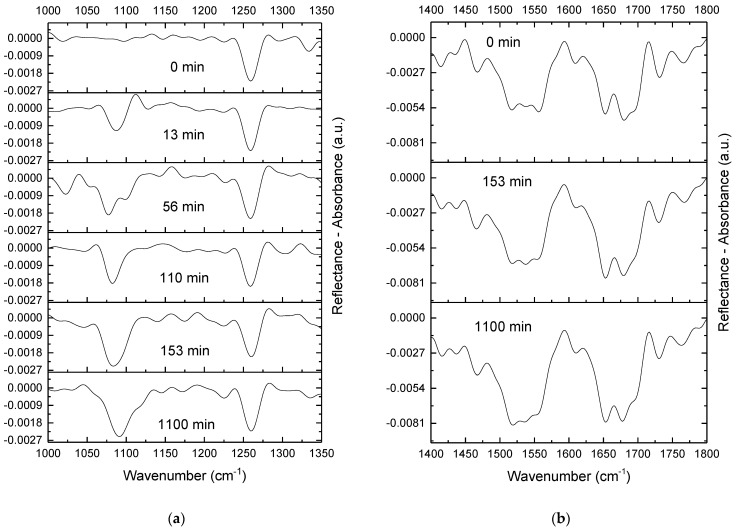
IRRAS spectra at 1000–1350 cm^−1^ (**a**) and 1400–1800 cm^−1^ (**b**) of a lysozyme film at different times after DNA injection into the subphase. The total DNA concentration is 50 µM.

**Figure 6 ijms-23-12377-f006:**
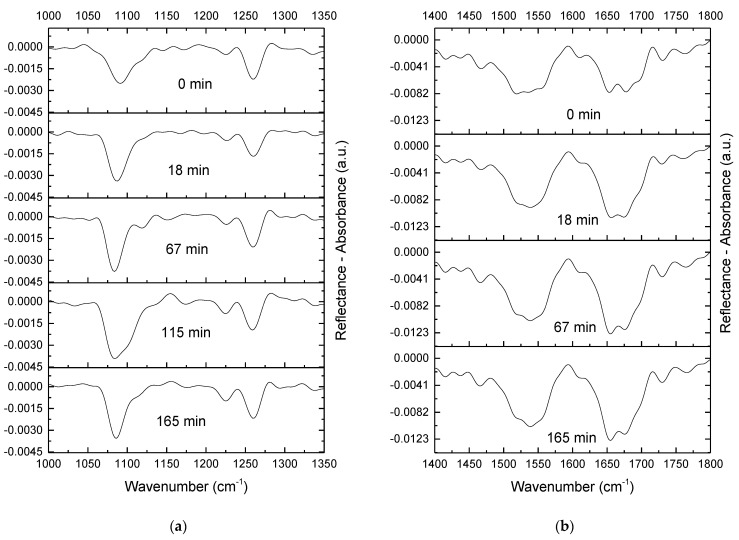
IRRAS spectra at 1000–1350 cm^−1^ (**a**) and 1400–1800 cm^−1^ (**b**) of a lysozyme film on a DNA-containing subphase at different times after the two-fold surface compression. The total DNA concentration is 50 µM.

## Data Availability

Data are contained within this article.

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
