# Peer review of "DNA Penetration into a Lysozyme Layer at the Surface of Aqueous Solutions"

_ijms, 2022, doi:10.3390/ijms232012377_

Round 1

Reviewer 1 Report

The authors adopted a series of techniques to characterize how DNA penetrate a lysozyme layer at the surface of aqueous solutions. The results clearly showed that DNA interacted with lysozyme and formed aggregates. The penetration process was faster than what was observed for DNA/PDAHMAC. The data was sufficient, and the manuscript was well written. However, I have a few concerns. 

My major concern is the novelty of the manuscript. The study seemed to be an extension for what has been studied for DNA/PDAHMAC, i.e., ref 19 in the manuscript. Given that lysozyme interacts weaker with DNA than the polymer, I wouldn’t be surprised of the current results. The authors could discuss the implications of the observations in our understanding of biology or engineering.

Minor concerns:

1. The authors didn’t describe clearly about what DNA were used. Were they random DNA with random length?

2. In Figure 5 and 6, the authors stated that “DNA injection into the subphase results in the emergence 225 of two bands at around 1085 and 1225 cm-1”. I could observe the 1085 band but was not convinced about the 1225 cm-1 one. Could the authors show clear evidence that the 1225 cm-1 band was significant?

Author Response

The authors adopted a series of techniques to characterize how DNA penetrate a lysozyme layer at the surface of aqueous solutions. The results clearly showed that DNA interacted with lysozyme and formed aggregates. The penetration process was faster than what was observed for DNA/PDAHMAC. The data was sufficient, and the manuscript was well written. However, I have a few concerns. 

My major concern is the novelty of the manuscript. The study seemed to be an extension for what has been studied for DNA/PDAHMAC, i.e., ref 19 in the manuscript. Given that lysozyme interacts weaker with DNA than the polymer, I wouldn’t be surprised of the current results. The authors could discuss the implications of the observations in our understanding of biology or engineering.

Authors' reply:

We are grateful to the reviewer for valuable remarks.

The reviewer is right that the electrostatic interactions in DNA/lysozyme complexes are presumably weaker than in DNA/PDAHMAC complexes. At the same time, this type of interactions does not entirely determine the morphology of the complexes. For example, the DNA/lysozyme complexes can form various aggregates in the bulk phase - rather flexible worm-like strands, compact rods, globules, and flower-like aggregates [Refs. 1,2 of this reply]. Aggregates of various shapes were also observed in mixed solutions of DNA and histones – the proteins, which have similar structures to lysozyme [Refs. 3,4 of this reply]. At relatively low surface concentrations (low surface pressures) we observed extended threads in a mixed DNA/lysozyme layer (Fig. 3b), which resemble the aggregates in DNA/PDAHMAC layers at intermediate surface lifetimes, but the layer morphology changes strongly at the increase of surface pressure. In contrast to the DNA/PDAHMAC layers we observed some folds and ridges (Figs. 3c and 3d). Such aggregates have never been observed in DNA/lysozyme solutions to the best of our knowledge. At the same time, the weaker electrostatic interactions as compared with DNA/PDAHMAC complexes may not be the only cause of the formation of different aggregates. It is possible that the high local lysozyme concentration in the layer induces the stabilization of non-canonical structures of DNA, for example, loops comprising unpaired nucleotides. These structures are quite probable in our polydisperse DNA samples and in this case the crowding of lysozyme results in strong interactions with them and not with duplex DNA structures [Refs. 5,6 of this reply]. The lysozyme interactions with loops of unpaired nucleotides do not favor the formation of long strands of a relatively small diameter but induce the formation of more disordered structures at high protein concentration. The high lysozyme concentration in spread layers mimics intracellular crowding and therefore the investigated system can model the effects of intracellular crowding on DNA behavior in living cells. We also believe that the obtained information on the formation of DNA/lysozyme aggregates in the surface layer will be useful for nanotechnology and nanoarchitectonics but it is too early to propose any specific applications. The following changes were inserted in the new version of the manuscript to clarify the novelty of the obtained results:     

“A special interest is generated by network structures of DNA/protein complexes, for example,  DNA/histone complexes, which can be formed upon adsorption onto the mica surface [9]. It has recently been demonstrated that such networks can possess antimicrobial properties, depending on their composition [10]. It is unclear, however, if such structures can be formed at the liquid – fluid interface.” (Page 2)

“The morphology of DNA-lysozyme aggregates depends strongly on the molar ratio between the components. Lundberg et al. observed the formation of DNA/lysozyme flexible worm-like assemblies at low lysozyme to DNA molar ratio [18]. Zhang et al. have recently demonstrated that binding of lysozyme globules to DNA can result in the formation of both “over-” and “undercharged” complexes with different morphology due to electrostatic interactions governed by the protein concentration [17]. Very recently Morimoto et al. studied the influence of high lysozyme concentrations, macromolecular crowding conditions, on it [19]. It proved that at high concentrations lysozyme stabilize mainly non-canonical structures of DNA, in particular, loops comprising unpaired nucleotides but not DNA duplexes.” (Page 2)

“Relatively weak interactions of lysozyme molecules with duplex DNA at high protein concentrations have been corroborated by a recent stability analysis of DNA oligonucleotide structures [19]. At the same time, it turned out that basic globular proteins at high concentrations stabilized strongly bulge loop structures. The non-canonical structures of DNA, for example, loops comprising unpaired nucleotides are quite probable in the investigated polydisperse DNA samples and in this case the crowding of lysozyme results in strong interactions with them and not with duplex DNA structures. The lysozyme interactions with loops of unpaired nucleotides presumably do not favor the formation of long strands of a relatively small diameter but induce the formation of more disordered structures at high protein concentration. The high lysozyme concentration in spread layers mimics intracellular crowding and therefore the investigated system can model the effects of intracellular crowding on DNA behavior in living cells.” (Page 10)

Minor concerns:

  1. The authors didn’t describe clearly about what DNA were used. Were they random DNA with random length?

Authors' reply

In our study, we used sodium salt of deoxyribonucleic acid from calf thymus acquired from Sigma-Aldrich (D1501) without further purification. This type of DNA is known to be highly polymerized and polydisperse [Ref. 7 of this reply]. However, it is often used to study the non-sequence-specific DNA interaction with various ligands in the bulk phase, as well as at the interface [Refs. 8–9 of this reply].

This information is now provided in Section 2.1. in the new version of the manuscript.

  1. In Figure 5 and 6, the authors stated that “DNA injection into the subphase results in the emergence 225 of two bands at around 1085 and 1225 cm-1”. I could observe the 1085 band but was not convinced about the 1225 cm-1 one. Could the authors show clear evidence that the 1225 cm-1 band was significant?

Authors' reply

1085 and 1225 cm-1 bands arise due to the phosphate moieties present in DNA and absent in lysozyme, indicating the DNA penetration into the lysozyme layer. At the same time, the intensity of the 1225 cm-1 band is much smaller than that of 1085 cm-1 band, so it is close to error limits before the surface compression (Figure 5a), but exceeds them a little after the compression (Figure 6a) due to an increase in the DNA surface concentration.

The following explanation was inserted in the new version of the manuscript:

“The intensity of the weak 1225 cm-1 band is much smaller than that of 1085 cm-1 band and is close to error limits before the surface compression (Figure 5a), but exceeds them a little after the compression (Figure 6a).”

References of this reply

  1. Lundberg, D.; Carnerup, A.M.; Janiak, J.; Schillén, K.; Miguel, M.D.G.; Lindman, B. Size and Morphology of Assemblies Formed by DNA and Lysozyme in Dilute Aqueous Mixtures. Phys. Chem. Chem. Phys. 2011, 13, 3082–3091.
  2. Zhang, R.; Wang, Y.; Yang, G. DNA–Lysozyme Nanoarchitectonics: Quantitative Investigation on Charge Inversion and Compaction. Polymers (Basel). 2022, 14, 1377.
  3. Liu, Y.Y.; Guthold, M.; Snyder, M.J.; Lu, H.F. AFM of Self-Assembled Lambda DNA-Histone Networks. Colloids Surfaces B Biointerfaces 2015, 134, 17–25.
  4. Song, Y.; Kadiyala, U.; Weerappuli, P.; Valdez, J.J.; Yalavarthi, S.; Louttit, C.; Knight, J.S.; Moon, J.J.; Weiss, D.S.; VanEpps, J.S.; Takayama, S. Antimicrobial Microwebs of DNA–Histone Inspired from Neutrophil Extracellular Traps. Adv. Mater. 2019, 31, 1807436.
  5. Nakano, S.I.; Miyoshi, D.; Sugimoto, N. Effects of Molecular Crowding on the Structures, Interactions, and Functions of Nucleic Acids. Chem. Rev. 2014, 114, 2733–2758.
  6. Morimoto, R.; Horita, M.; Yamaguchi, D.; Nakai, H.; Nakano, S. ichi Evaluation of Weak Interactions of Proteins and Organic Cations with DNA Duplex Structures. Biophys. J. 2022, 121, 2873–2881.
  7. Porsch, B.; Laga, R.; Horský, J.; Koňák, ÄŒ.; Ulbrich, K. Molecular Weight and Polydispersity of Calf-Thymus DNA: Static Light-Scattering and Size-Exclusion Chromatography with Dual Detection. Biomacromolecules 2009, 10, 3148–3150.
  8. Maldonado-Valderrama, J.; Yang, Y.; Jiménez-Guerra, M.; Del Castillo-Santaella, T.; Ramos, J.; Martín-Molina, A. Complexation of DNA with Thermoresponsive Charged Microgels: Role of Swelling State and Electrostatics. Gels 2022, 8, 184.
  9. Janich, C.; Hädicke, A.; Bakowsky, U.; Brezesinski, G.; Wölk, C. Interaction of DNA with Cationic Lipid Mixtures - Investigation at Langmuir Lipid Monolayers. Langmuir 2017, 33, 10172–10183.

Reviewer 2 Report

This article written by an international team of authors is of interest, since the topic of DNA has been relevant for many years. During this time, a lot of interesting information on this topic has appeared in the literature. The advantage of this work is that the authors managed to take a fresh look at this topic and make their contribution. The article is well written and understandable. However, there are things to fix:

1. The abstract can be extended.

2. I strongly recommend that authors make more comparisons with literary sources. This will add more validity to the conclusions.

3. In the introduction, please specify the purpose.

4. The conclusions are also written very modestly.

5. It would be good if additional methods of physical and chemical analysis were added.

Author Response

This article written by an international team of authors is of interest, since the topic of DNA has been relevant for many years. During this time, a lot of interesting information on this topic has appeared in the literature. The advantage of this work is that the authors managed to take a fresh look at this topic and make their contribution. The article is well written and understandable. However, there are things to fix:

  1. The abstract can be extended.

Authors' reply

We are grateful to the reviewer for the remarks and changed our manuscript according to them.

The abstract has been extended in the new version of the manuscript. The following text was added to the abstract:

“At low surface pressures the AFM images show the formation of long strands in the surface layer. The further surface compression does not lead to the formation of a network of DNA/lysozyme aggregates as in the case of a mixed layer of DNA and synthetic polyelectolytes but to the appearance of some folds and ridges in the layer, which can be caused by the strong interaction of lysozyme with loops of unpaired nucleotides in a crowded molecular environment.”   

  1. I strongly recommend that authors make more comparisons with literary sources. This will add more validity to the conclusions.

Authors' reply

More comparisons with literary sources were added to the new version of the manuscript:

“A special interest is generated by network structures of DNA/protein complexes, for example,  DNA/histone complexes, which can be formed upon adsorption onto the mica surface [9]. It has recently been demonstrated that such networks can possess antimicrobial properties, depending on their composition [10]. It is unclear, however, if such structures can be formed at the liquid – fluid interface.” (Page 2)

“The morphology of DNA-lysozyme aggregates depends strongly on the molar ratio between the components. Lundberg et al. observed the formation of DNA/lysozyme flexible worm-like assemblies at low lysozyme to DNA molar ratio [18]. Zhang et al. have recently demonstrated that binding of lysozyme globules to DNA can result in the formation of both “over-” and “undercharged” complexes with different morphology due to electrostatic interactions governed by the protein concentration [17]. Very recently Morimoto et al. studied the influence of high lysozyme concentrations, macromolecular crowding conditions, on it [19]. It proved that at high concentrations lysozyme stabilize mainly non-canonical structures of DNA, in particular, loops comprising unpaired nucleotides but not DNA duplexes.” (Page 2)

“Relatively weak interactions of lysozyme molecules with duplex DNA at high protein concentrations have been corroborated by a recent stability analysis of DNA oligonucleotide structures [19]. At the same time, it turned out that basic globular proteins at high concentrations stabilized strongly bulge loop structures. The non-canonical structures of DNA, for example, loops comprising unpaired nucleotides are quite probable in the investigated polydisperse DNA samples and in this case the crowding of lysozyme results in strong interactions with them and not with duplex DNA structures. The lysozyme interactions with loops of unpaired nucleotides presumably do not favor the formation of long strands of a relatively small diameter but induce the formation of more disordered structures at high protein concentration. The high lysozyme concentration in spread layers mimics intracellular crowding and therefore the investigated system can model the effects of intracellular crowding on DNA behavior in living cells.” (Page 10)

  1. In the introduction, please specify the purpose.

Authors' reply

The purposes are now specified in the new version of the manuscript:

“The main aim is to elucidate the peculiarities of DNA/protein interactions at the liquid – gas interface and to check the possibility of a network formation. A special purpose is to estimate the DNA/lysozyme aggregation in a crowded molecular environment in the lysozyme spread layer.” (Page 2)

  1. The conclusions are also written very modestly.

Authors reply

The conclusions section was expanded with following sentences:

“The observed peculiarities of the properties of mixed DNA/lysozyme layers have presumably two main causes. First, the interactions between the protein and lysozyme layer are weaker than in the case of DNA penetration into PDAHMAC layers. Second, lysozyme interacts at high local concentrations in the surface layer mainly with non-canonical structures of DNA leading to their stabilization and the formation of more disordered aggregates.” (Page 12)

  1. It would be good if additional methods of physical and chemical analysis were added.

Authors' reply

We agree with the reviewer that the application of additional methods of physical and chemical analysis would be extremely useful. Unfortunately, the number of available techniques in the surface chemistry of liquids is quite limited. At the moment we are trying to use neutron reflectometry and applied for the beam time in the Joint Institute for Nuclear Research in Dubna (Moscow region). The application procedures are rather lengthy and it is difficult to predict when we will be able to use this equipment.

Round 2

Reviewer 1 Report

The authors have address my concerns adequately.